# Performance Evaluation of CentiSpace Navigation Augmentation Experiment Satellites

**DOI:** 10.3390/s23125704

**Published:** 2023-06-19

**Authors:** Lin Chen, Feiren Lv, Qiangwen Yang, Tulin Xiong, Yuqi Liu, Yi Yang, Hongchen Pan, Suisheng Wang, Min Liu, Renlun He, Duo Zheng, Lingzhi Zhang, Yundi Jin

**Affiliations:** 1The 29th Research Institute of China Electronic Technology Group Corporation, Chengdu 610036, China; 2Beijing Future Navigation Tech Co., Ltd., Beijing 100081, China

**Keywords:** LEO navigation augmentation, self-interference suppression, GNSS receiver, signal quality

## Abstract

This paper presents the performance analysis of CentiSpace low earth orbit (LEO) experiment satellites. Distinguishing them from other LEO navigation augmentation systems, the co-time and co-frequency (CCST) self-interference suppression technique is employed in CentiSpace to mitigate significant self-interference caused by augmentation signals. Consequently, CentiSpace exhibits the capability of receiving navigation signals from the Global Navigation Satellite System (GNSS) while simultaneously broadcasting augmentation signals within the same frequency bands, thus ensuring excellent compatibility for GNSS receivers. CentiSpace is a pioneering LEO navigation system to successfully complete in-orbit verification of this technique. Leveraging the on-board experiment data, this study analyzes the performance of space-borne GNSS receivers equipped with self-interference suppression and evaluates the quality of navigation augmentation signals. The results show that CentiSpace space-borne GNSS receivers are capable of covering more than 90% visible GNSS satellites and the precision of self-orbit determination is at the centimeter level. Furthermore, the quality of augmentation signals meets the requirements outlined in the BDS interface control documents. These findings underscore the potential of the CentiSpace LEO augmentation system for the establishment of global integrity monitoring and GNSS signal augmentation. Moreover, these results contribute to subsequent research on LEO augmentation techniques.

## 1. Introduction

The BeiDou navigation satellite system (BDS) establishes and maintains unified space–time reference within its coverage. BDS provides positioning, navigation and timing (PNT) service for various users, which meets the most common needs. With the increasing demand for high-precision, reliable and global PNT service, the construction of the next navigation satellite system has become a hot topic in the field of navigation technology [1]. Due to the global multiple coverage for GNSS satellites, rapid geometric configuration change and high signal landing power, the LEO satellite system is expected to become an important part of the next navigation satellite system [2].

Extensive studies have been carried out on the LEO satellite system. Iridium and SpaceX constellations in the United States and the Sphere/Sfera constellation in Russia proposed navigation augmentation schemes with communication signals [3,4]. The China Satellite Network Group Co., Ltd. is also officially starting the construction of a communication and navigation fusion system. Luojia-1 and Tianshu-1 satellites in China employed dedicated navigation platforms and carried out augmentation research with navigation signals.

CentiSpace was built by BeiJing Future Navigation Technology Company in collaboration with The 29th Research Institute of China Electronic Technology Group Corporation (CETC-29). As early as September 2018, the CentiSpace navigation augmentation system launched the first experiment satellite, completing the preliminary signal verification. From September to December 2022, four CentiSpace experiment satellites were launched for further research, such as GNSS integrity monitoring, autonomous orbit determination and navigation augmentation. Different from other LEO satellite systems, CentiSpace satellites not only receive navigation signals from GNSS but also broadcast augmentation signals (in GNSS frequency bands) at the same time. Such capability provides compatibility of global integrity monitoring and GNSS signal augmentation.

The navigation augmentation service of CentiSpace includes global integrity monitoring and GNSS signal augmentation. Navigation augmentation equipment is composed of space-borne GNSS receivers and augmentation signal payloads. Compared to ground monitoring networks, CentiSpace space-borne GNSS receivers can perform multiple monitoring of GNSS satellites and are less affected by atmospheric delay and multipath interference, which contributes to providing global high-precision integrity service and improves the safety and reliability of PNT services. On the other hand, the geometric configuration change of LEO satellites is 20 times that of medium earth orbit (MEO) satellites, which reduces the coherence between observation epochs. So the augmentation signals are able to accelerate the convergence time of high-precision positioning [5]. Augmentation signal payloads of CentiSpace broadcast FA signals and FB signals in GNSS frequency bands which is near B1 frequency and B2 frequency, respectively, as shown in Table 1. The modulation method of the FA and FB signals is BPSK, and the code rate is 2.046 MHz.

This paper aims to evaluate the performance of space-borne GNSS receivers with self-interference and the quality of augmentation signals for CentiSpace experiment satellites. Distinguished from prior literature, this study presents a comprehensive analysis of LEO practical performance pertaining to both the signal reception and transmission with identical frequency. It establishes a robust data foundation to advance the theoretical investigation of LEO navigation augmentation. The results of GNSS receivers are also the first in-orbit verification of the CCST technique in LEO satellites, which will expand the potential for LEO navigation augmentation. The methods of evaluation are described in Section 2, followed by the evaluation results in Section 3. Finally, conclusions are presented.

## 2. Performance Evaluation Methods

CentiSpace experiment satellites receive navigation signals while broadcasting augmentation signals in the same frequency bands. Thus, it is imperative to conduct a synchronized analysis of GNSS receivers, navigation augmentation signals and the reciprocal influence between the two components. Figure 1 shows the framework diagram of the CentiSpace performance evaluation.

### 2.1. GNSS Receiver Evaluation Methods

The primary contribution of CentiSpace GNSS receivers is integrity monitoring, so it is necessary to analyze the integrity of raw measurements. Additionally, CentiSpace GNSS receivers provide an autonomous orbit determination service. The main evaluation indices for this service include the assessment of multipath and observation noise error [6].

Different from common receiving terminals, frequency points of the navigation augmentation signals are close to GNSS frequency bands (as shown in Table 1), which brings strong co-frequency self-interference for GNSS receivers. Self-interference suppression technique in co-time and co-frequency is applied to solve this problem. Therefore, the variation of C/N0 within and without co-frequency self-interference is also an important index to characterize the performance of CentiSpace GNSS receivers.

In summary, integrity of raw measurements, the variation of C/N0 within and without co-frequency self-interference, multipath error, observation noise error and orbit determination precision are mainly analyzed below.

#### 2.1.1. Integrity of Raw Measurements

The integrity of raw measurements is defined as the ratio of the real dual-frequency epoch number to the theoretical epoch number. Due to series problems such as shadowing effect, signals tracked by GNSS receivers may be discontinuous, which causes the index to be less than 100%. So the index characterizes the receiving ability of GNSS receivers.

The evaluation method of the index is to count the number of raw measurements during the evaluation periods and compare it with the number of theory. Then, the integrity rates of all visible GNSS satellites are statistically averaged as the evaluation result. The calculation method is shown in the following formula:(1)$$\left\{\begin{array}{c} C_{i}=\frac{N_{i}^{r}}{N_{i}^{t}}\\ I=\frac{\sum_{i=1}^{n}C_{i}}{n} \end{array}\right.$$
where character $C_{i}$ is the integrity of the *i*th GNSS satellite, $N_{i}^{t}$ is the theoretical number and $N_{i}^{r}$ is the real number. *I* is the integrity of raw measurements of the GNSS receiver. Character *n* represents the total number of GNSS satellites received. The strategy to accept a raw measurement ($R_{m}$) is shown in Figure 2. It is essential to emphasize that the data error must satisfy specific conditions, as indicated by the “Value Check” depicted in the diagram.

#### 2.1.2. Variation of C/N0 with Co-Frequency Self-Interference

The power of CentiSpace augmentation signals is more than 100 dB higher than that of the normal signals received by the antenna. Despite the spatial separation and opposite pointing directions of the transmitting and receiving antennas on the satellite, this still results in a significant increase in noise levels that can impair the signal receiving ability of CentiSpace GNSS receivers. The variation of C/N0 with co-frequency self-interference is evaluated in this paper, which directly characterizes the self-interference suppression performance of CentiSpace GNSS receivers.

The theoretical variation curve of C/N0 is obtained by high-order fitting of the C/N0 without co-frequency interference. Then, C/N0 with co-frequency self-interference is compared with the theoretical C/N0 and the influence of co-frequency self-interference is obtained. The variation of C/N0 with co-frequency self-interference is checked by the statistical average results of multiple satellites.

#### 2.1.3. Multipath Error

The multipath effect refers to the interference caused by multipath signals during propagation. The working environment is relatively pure for space-borne GNSS receivers and the main multipath signals come from the reflection of solar panels. According to the relevant research, the multipath error on the carrier phase is less than 1/4 wavelength of the carrier, while the pseudo-range multipath error is 200 times that of the carrier multipath error [7]. Therefore, the multipath analysis object is the pseudo-range multipath error in this paper.

The evaluation method mainly depends on the dual-frequency observation data. MP combinations are applied to evaluate the pseudo-range multipath error [7], as shown in the following formula:(2)$$\left\{\begin{array}{c} MP1=\rho_{1}-\lambda_{1}\phi_{1}\frac{f_{1}^{2}+f_{2}^{2}}{f_{1}^{2}-f_{2}^{2}}+\lambda_{2}\phi_{2}\frac{2f_{2}^{2}}{f_{1}^{2}-f_{2}^{2}}\\ MP2=\rho_{2}-\lambda_{1}\phi_{1}\frac{2f_{1}^{2}}{f_{1}^{2}-f_{2}^{2}}+\lambda_{2}\phi_{2}\frac{f_{1}^{2}+f_{2}^{2}}{f_{1}^{2}-f_{2}^{2}} \end{array}\right.$$
where ρ is the pseudo-range, λ is the wavelength, ϕ is the carrier phase and *f* is the frequency point. The statistical evaluation value of pseudo-range multipath error can be obtained by smoothing the MP combinations:(3)$$\left\{\begin{array}{c} V1_{k}=\sqrt{\frac{1}{n-1}\sum_{k}^{k+n-1}{\left(MP1_{k}-\frac{\sum_{k}^{k+n-1}MP1_{k}}{n}\right)}^{2}}\\ V2_{k}=\sqrt{\frac{1}{n-1}\sum_{k}^{k+n-1}{\left(MP2_{k}-\frac{\sum_{k}^{k+n-1}MP2_{k}}{n}\right)}^{2}} \end{array}\right.$$
where *n* denotes the size of the sliding window, and $V1_{k}$ and $V2_{k}$ are evaluation results.

#### 2.1.4. Observation Noise Error

Observation noise error refers to the pseudo-range and carrier phase measurement deviation. It is influenced by measurement noise, incompletely eliminated atmospheric delay error, modeled satellite orbit error, clock error, etc. The evaluation methods mainly include polynomial fitting and high-order difference. The polynomial fitting method calculates high-order linear fitting on the observation epochs. The high-order difference method calculates the high-order difference between the epochs. The method also calculates the root mean square of the high-order difference and eliminates the amplification effect of the difference [6].

The polynomial fitting method requires high epoch sampling rate. Considering the communication capacity of data transmission link, this paper uses high-order difference method to evaluate the observation noise. The evaluation method is described as Formula (Equation 4). If the three-order difference method is used, then the result is illustrated as Formula (Equation 5). In the formula, *O* represents raw measurements array; *E* denotes the mathematical expectation. *K* means the amplification effect of the difference; σ is the evaluation result of the observation noise.
(4)$$\left\{\begin{array}{c} \Delta O_{i}=O_{i}-O_{i-1}\\ \Delta \Delta O_{i}=\Delta O_{i}-\Delta O_{i-1}\\ \Delta \Delta \Delta O_{i}=\Delta \Delta O_{i}-\Delta \Delta O_{i-1}\\ \ldots \ldots \end{array}\right.$$
(5)$$\left\{\begin{array}{c} \sigma_{0}=\frac{1}{n-1}\sum_{i=1}^{n}{\left[\Delta \Delta \Delta O_{i}-E\left(\Delta \Delta \Delta O_{i}\right)\right]}^{2}\\ \sigma =\frac{\sigma_{0}}{K} \end{array}\right.$$

#### 2.1.5. Orbit Determination

Based on raw measurements provided by space-borne GNSS receivers, the orbit parameters of LEO satellites can be estimated with the known kinematic laws. The data processing is shown in Figure 3. In the re-processing step, dual-frequency ionosphere-free combinations are used and positions of GNSS satellites are also calculated with GNSS ephemeris.

The precision of orbit determination is influenced by raw measurements, GNSS ephemeris and orbit dynamic model. In order to improve the precision of orbit determination, the pseudo-range observation noise error is usually required to be less than 30 cm and the constraint for carrier phase is less than 2 mm [8]. With the precise ephemeris released by International GNSS Service (IGS), we can reduce the influence of GNSS satellite position error.

### 2.2. Augmentation Signal Evaluation Methods

CentiSpace navigation augmentation system provides users with signal and information enhancements. Subscribers can easily receive augmentation signals through simple terminals and directly perceive and evaluate the navigation augmentation services. Monitoring and evaluating the quality of these augmentation signals is therefore crucial to ensure the stability of the system and provide subscribers with excellent services.

The monitoring and evaluation equipment for CentiSpace navigation augmentation signals is composed of parabolic antenna, signal filter, low noise amplifier, signal collector and quality analysis software, as shown in Figure 4. The aperture of parabolic antenna is 7.4 m, which provides about 30 dBi antenna gain for the receiving of augmentation signals. Signal quality analysis software processes the collected signals by downconversion, acquisition, tracking, etc. The quality of navigation augmentation signals in modulation domain and correlation domain is analyzed in this paper.

#### 2.2.1. Quality Analysis in Modulation Domain

In this paper, distortion degree of orthogonality is analyzed which represents the signal quality in modulation domain. The expression of the signal is shown in the following formula, where *A* is the amplitude of the signal, *I* and *Q* represents in-phase and quadrature signals, respectively, *C* is the pseudo-codes, *D* is the navigation augmentation message and ϕ is the phase of signal.
(6)$$S\left(t\right)=A_{I}C_{I}cos\left(2\pi ft+\phi \right)+A_{Q}C_{Q}Dsin\left(2\pi ft+\phi \right)$$

Theoretically, the in-phase and quadrature signal are orthogonal. Due to the the non-idealization of power amplifier components and transmission channels, the orthogonality of in-phase and quadrature signal may be reduced. This orthogonal error leads to the tracking deviation of carrier phase and pseudo-codes, which will affect the high-precision positioning performance [9].

In the quality analysis software, the received signals are acquired and tracked, and we can rebuild the ideal in-phase and quadrature baseband signals. Then the received signals (after downconversion) are correlated with the ideal baseband signals and the maximum correlation peaks are obtained. The phases at the maximum correlation peak are also calculated which are regarded as the phases of the in-phase and quadrature signals, respectively. Lastly, the distortion degree of orthogonality is the phase difference of in-phase and quadrature signals.

#### 2.2.2. Quality Analysis in Correlation Domain

Subjected to the non-idealization of core devices on-board such as frequency converters, filters and high-power amplifiers, the cross-correlation peak of augmentation signal is reduced and the shape of the cross-correlation function is deteriorated. It is necessary to analyze the correlation loss and zero-crossing bias of S-curve for the quality analysis in correlation domain.

(1)Correlation loss

Correlation loss is defined as the attenuation of the real correlation to ideal correlation. The correlation loss reflects the proportion of useful signals contained in the received signal and the reduction degree of correlation peaks. In addition, the present branch is used in carrier tracking loop to complete the phase discrimination, so the correlation function also characterizes the effective C/N0 attenuation caused by signal distortion [10].

The evaluation method of correlation loss is shown in the following formula. In the formula, CCF represents the correlation function, *P* represents the signal power, and $S_{ref}\left(t\right)$ and $S_{rec}\left(t\right)$ represent the received signals and the local reference signals, respectively. The local reference signals are built by the results of tracking.
(7)$$L\left(dB\right)=20\log_{10}\frac{\max_{\tau }CCF(S_{rec}\left(t-\tau \right),S_{ref}\left(t\right))}{\sqrt{P(S_{rec}\left(t\right))}\times \sqrt{P(S_{rec}\left(t\right))}}$$

(2)Zero-Crossing Bias of S-Curve

Zero-crossing bias of S-curve characterizes the asymmetry of the correlation function between the actual received signals and the ideal signals. The distortion of the received signal leads to the asymmetry of the correlation function, and there will be a deviation between the code phase and the real code phase, even if the output of the code discriminator is zero [11]. Zero-crossing bias of S-curve is the quantitative result of this deviation.

In this paper. EMLP discriminator is used for the code phase discriminator [12]. The phase discriminator function is shown in the following formula, where δ is the interval of correlator.
(8)$$\begin{array}{c} D\left(\tau ,\delta \right)=\;|CCF\left(S_{rec}\left(S_{ref}\left(t\right),t+\tau +\frac{\delta }{2}\right)\right){|}^{2}\\ -|CCF\left(S_{rec}\left(S_{ref}\left(t\right),t+\tau -\frac{\delta }{2}\right)\right){|}^{2} \end{array}$$

$\tau_{0}$ is the theoretical data when CCF gets the maximum value. If $\tau_{p}$ satisfies $D(\tau_{p},\delta )=0$, then zero-crossing bias of S-curve is calculated as $\tau_{p}-\tau_{0}$.

## 3. Performance of Experiment Satellite

CentiSpace navigation augmentation experiments are currently implemented in orbit. Based on the evaluation equipment and the the telemetry data of the space-borne GNSS receivers provided by CETC-29, the performance of CentiSpace navigation augmentation experiment satellites is presented and analyzed in this section.

### 3.1. Performance of GNSS Receiver

#### 3.1.1. Integrity of Raw Measurements

Considering the influence of observation quality, the epochs with elevation greater than 10° are selected for statistics. In addition, CentiSpace satellites broadcast augmentation signals in the GNSS band which may influence the GNSS receivers. So the integrity results within and without co-frequency self-interference are also analyzed.

CentiSpace space-borne GNSS receivers are able to cover more than 90% visible GNSS satellites. As an example, Figure 5 shows the integrity results of BDS dual-frequency signals. When the navigation augmentation signals are turned off, the integrity result is 91.25%, while the result with co-frequency self-interference is still 90.86%. The results also show that the integrity of CentiSpace GNSS receivers is not affected by the co-frequency self-interference signals through the self-interference suppression technique in co-time and co-frequency.

#### 3.1.2. Variation of C/N0 with Co-Frequency Self-Interference

Navigation augmentation signals may raise the noise level and affect the receiving ability of CentiSpace GNSS receivers. The variation of C/N0 with co-frequency self-interference is evaluated in this section, which directly characterizes the self-interference suppression technique. The frequency points of the navigation augmentation signals are near the B1 and B2 frequencies, respectively, so the variation of C/N0 for BDS B1C and B2a signals are chosen to assess the self-interference suppression technique. In the event of deactivating the interference suppression algorithm, GNSS receivers will experience signal disengagement. Figure 6 shows the variation of C/N0 for BDS B1C and B2a signals. As a result, the variation of C/N0 is less than 1dB in statistics, which is not affected by the co-frequency self-interference signals through the self-interference cancellation technique in co-time and co-frequency.

#### 3.1.3. Multipath Error

Due to the rotation of the solar panel, the solar panel may cause multipath components in the received signals. After cycle jump detection, pseudo-range multipath error receivers can be evaluated by MP combination. Figure 7 shows the MP combination for BDS C39. With the change in elevation, the evaluation results for the multipath error are 0.042–0.345 m for the B1C signal and 0.023–0.201 m for the B2a signal. The statistical results of all the BDS satellites are 0.027–0.482 m for the B1C signal and 0.019–0.332 m for the B2a signal, which are less than 0.5m and match the results from the iGMAS tracking stations [13].

#### 3.1.4. Observation Noise Error

In this paper, the high-order difference method is used to evaluate the observation noise error of the pseudo-range and carrier phases for CentiSpace GNSS receivers. The high-order difference results of BDS B1C and B2a signals without co-frequency self-interference are illustrated in Figure 8. When the navigation augmentation signals are turned on, the high-order difference results are shown in Figure 9. Different colors of line represent different satellites.

According to these figures, the observation noise error with low elevation (circled in Figure 8 and Figure 9) is greater than that with high elevation, especially for the carrier phase. So the data with elevation angle lower than 10° is also selected for statistic analysis.

The μ detection method is used for normality analysis; the verification data are described as the following formula, where *S* is the skewness, *K* is the kurtosis.
(9)$$\left\{\begin{array}{c} \mu_{1}=\frac{S}{\sqrt{6/n}}\\ \mu_{2}=\frac{K}{\sqrt{24/n}} \end{array}\right.$$

At the 95% confidence level, the values of $\mu_{1}$ and $\mu_{2}$ are in the range of [−2,2]. Therefore, the high-order difference method can reflect the random characteristics of raw measurements and the statistical characteristics of the normal distribution will be used to evaluate the observation noise of CentiSpace GNSS receivers.

The statistical results are shown in the following Table 2. Comparing the observation noise when the navigation augmentation signal is turned on and off, the results are not affected by the co-frequency self-interference signals through the self-interference suppression technique in co-time and co-frequency. The pseudo-range noise error of the space-borne GNSS receiver is within 80 mm and the carrier phase noise error is less than 2 mm, which can provide support for orbit determination of low-orbit satellites.

#### 3.1.5. Orbit Determination

The orbit determination scheme is described in Table 3 and the detailed satellite dynamics model is shown in [14]. The observation noise error (as mentioned above) is 80 mm for pseudo-range and 2 mm for carrier, which will help to improve the precision of the orbit determination. Dual-frequency combinations are used to suppress the atmosphere delay error and the precise ephemeris (from IGS) will improve the precision of GNSS satellites positions.

The orbit determination results are shown in Table 4. With separate BDS raw measurements, the accuracy in radial (R), transverse (T) and normal (N) is 1.05 cm, 2.06 cm and 1.84 cm, respectively. The accuracy will improve to 0.89 cm (R), 2.35 cm (T) and 1.26 cm (N) through multiple GNSS systems. The orbit determination precision is at the centimeter level.

### 3.2. Performance of Navigation Augmentation Signal

BDS interface control documents [15,16] specify the quality requirements of the B1C and B2a signals. The indexes are shown in Table 5. As shown in the table, the maximum values of correlation loss are 0.3 dB for the B1C signal and 0.6 dB for the B2a signal. The documents have not definitely stipulated the requirements of the zero-crossing bias of S-curve. The results of the reference [10] show that maximum values of zero-crossing bias of S-curve are near 0.3 ns for the B1C and B2a signals when the correlator interval is 1 chip. Performances of navigation augmentation signals are presented and compared with these requirements below.

#### 3.2.1. Quality analysis in Modulation Domain

In order to analyze the orthogonality of the in-phase and quadrature signals, the FA and FB signals are collected at different elevations. The phase relations (mod 90°) between in-phase and quadrature signals are shown in Table 6. The evaluation results of orthogonality are less than 0.5° at high and medium elevations, while the results are deteriorated at low elevation, which is because of the shadowing and ability of the receiving parabolic antenna (high elevation: from 60° to 90°, medium elevation: from 30° to 60°, low elevation: from 10° to 30°).

#### 3.2.2. Quality Analysis in Correlation Domain

Subjected to the non-idealization of core devices onboard, the cross-correlation peak of the augmentation signal is reduced and the shape of the cross-correlation function is deteriorated. According to the signal quality evaluation methods in Section 2, the correlation domain signal quality of the FA and FB signals is performed below.

(1)Correlation loss

The reduction degree of the correlation peak is quantified by correlation loss. The correlation loss of the FA and FB signals is illustrated in Table 7. The evaluation results of correlation loss are less than 0.3 dB at high and medium elevations, while results are over 0.3 dB at low elevations. According to the analysis, the C/N0 is decreased heavily compared with the data at medium elevations which causes the deterioration of correlation loss.

(2)Zero-Crossing Bias of S-Curve

Zero-crossing bias of S-curve characterizes the asymmetry of the correlation function between real received signals and ideal signals. Figure 10 shows the s-curve bias of FA and FB signal at different elevations. The results of the s-curve in different elevations are less than 0.3 ns when the correlator interval is 1 chip and the results are improved with higher elevations which is because of the increasing of the signal landing power.

## 4. Conclusions

This paper analyzes the performance of GNSS receivers and the quality of navigation augmentation signals based on the data from the CentiSpace experiment satellites. It is also the first in-orbit verification of the CCST technique in LEO satellites. The results show that CentiSpace GNSS receivers are able to cover more than 90% of visible GNSS satellites even with self-interference and the self-orbit determination precision is at the centimeter level. The analysis of augmentation signals also indicates that the signal quality of navigation augmentation signals is comparable to that of BDS-3 satellites.

CentiSpace is currently engaged in the endeavor to establish a global network comprising LEO satellites. Comprehensive investigations and validations pertaining to global integrity enhancement and high-precision positioning will be carried out later. The empirical data presented in this article holds potential significance as a point of reference for subsequent studies in the foreseeable future.

## Figures and Tables

**Figure 1 sensors-23-05704-f001:**
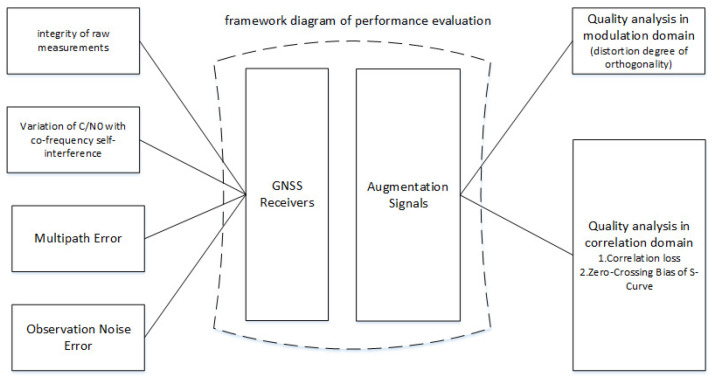
Framework diagram of CentiSpace performance evaluation.

**Figure 2 sensors-23-05704-f002:**
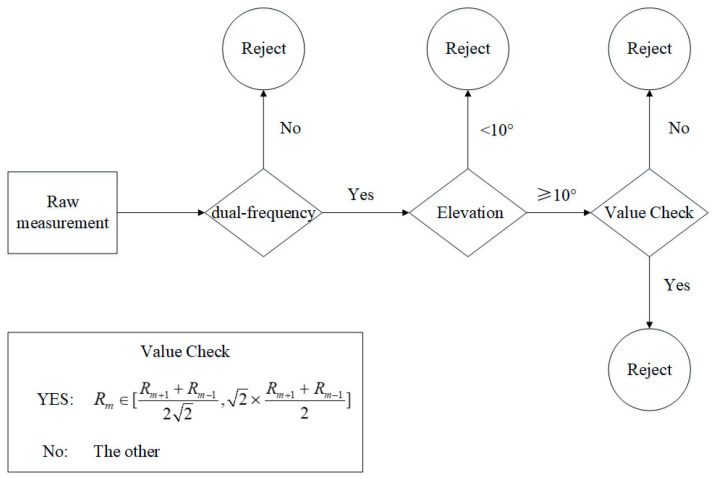
Strategy to accept a raw measurement.

**Figure 3 sensors-23-05704-f003:**
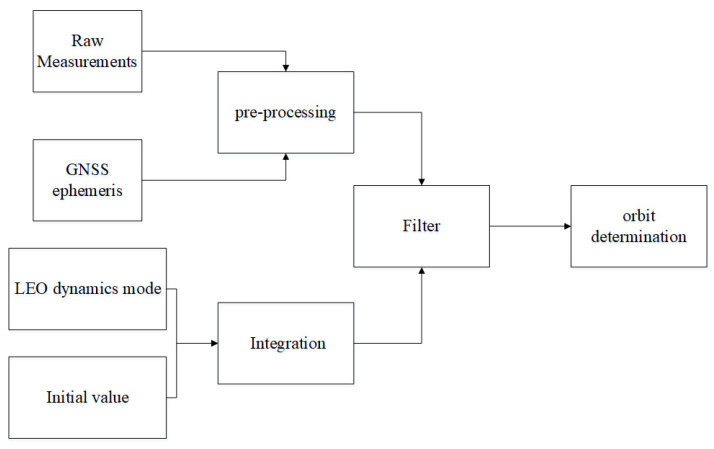
Orbit determination.

**Figure 4 sensors-23-05704-f004:**
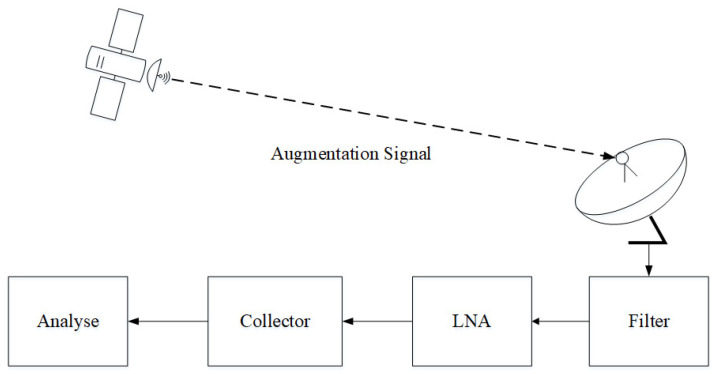
Signal evaluation system.

**Figure 5 sensors-23-05704-f005:**
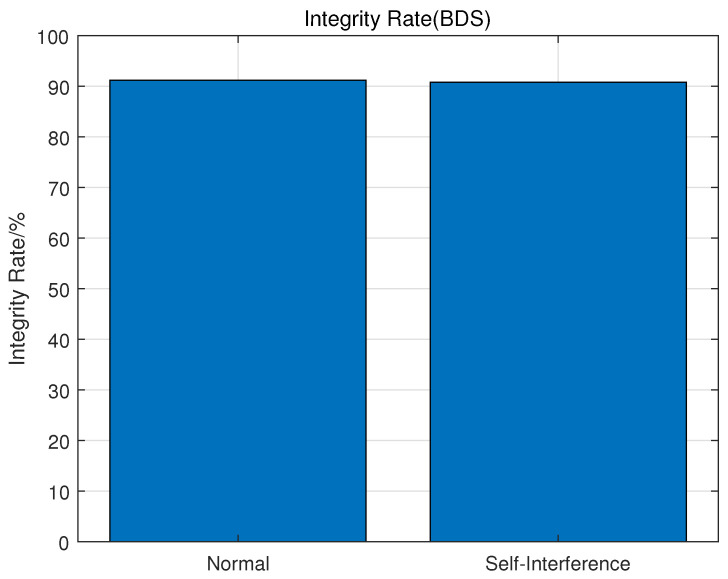
Integrity of BDS raw measurements.

**Figure 6 sensors-23-05704-f006:**
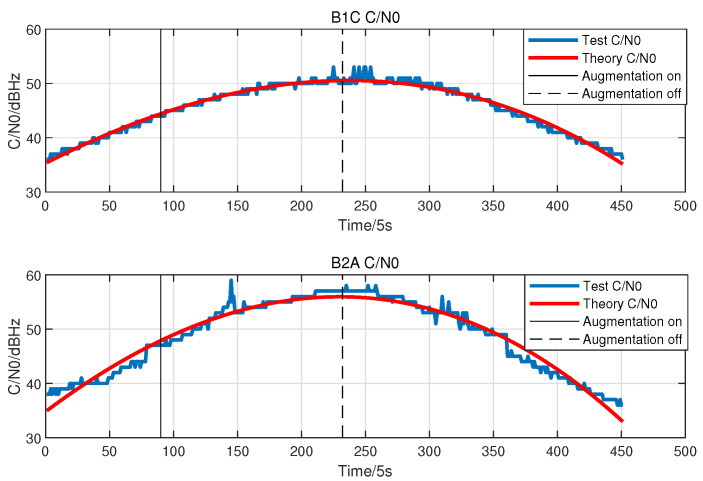
C/N0 variation with co-frequency self-interference.

**Figure 7 sensors-23-05704-f007:**
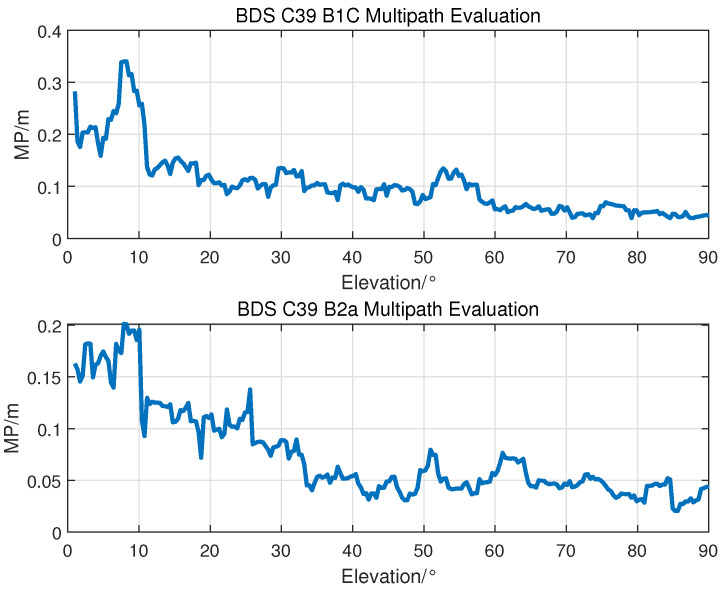
Multipath evaluation for BDS C39.

**Figure 8 sensors-23-05704-f008:**
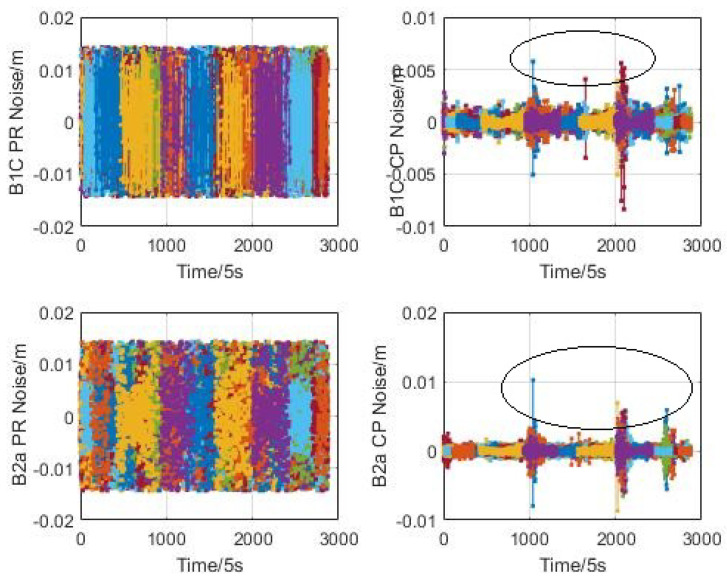
Augmentation signal off.

**Figure 9 sensors-23-05704-f009:**
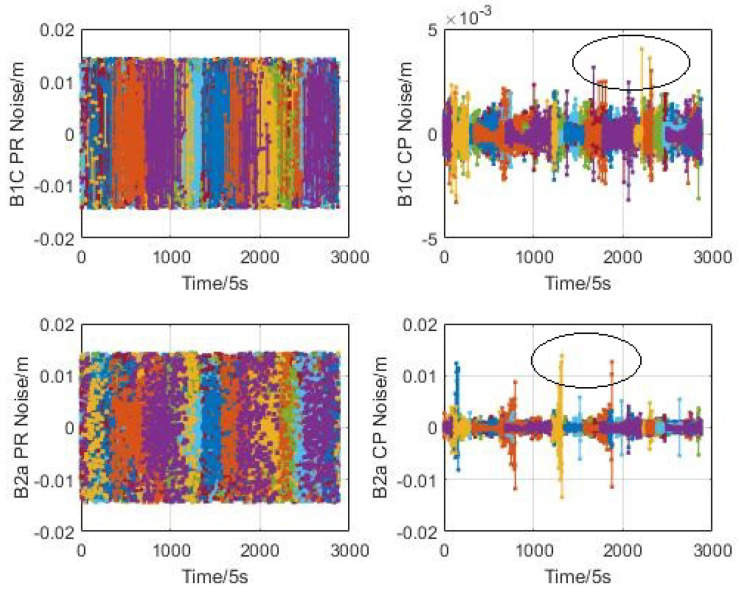
Augmentation signal on.

**Figure 10 sensors-23-05704-f010:**
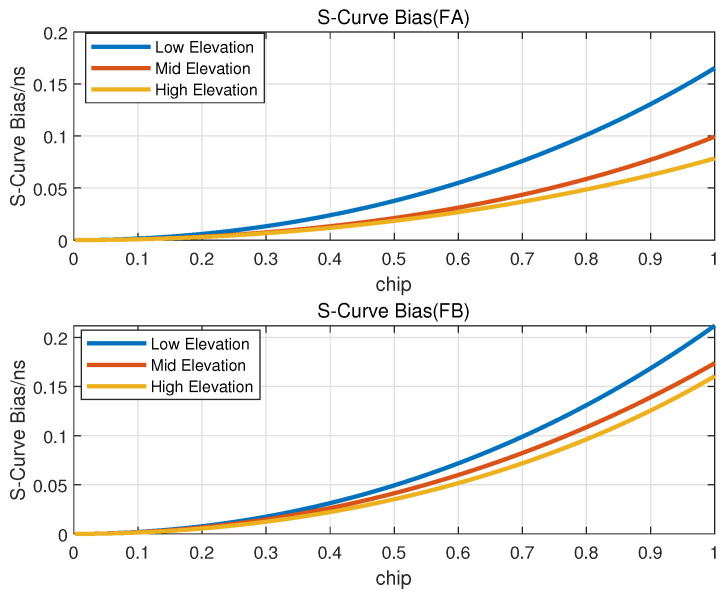
S-curve of of navigation augmentation signal.

**Table 1 sensors-23-05704-t001:** Navigation augmentation signal structure.

Signal	Frequency	Modulation	Code Rate
FA	157X.XX	BPSK	2.046 MHz
FB	117X.XX	BPSK	2.046 MHz

**Table 2 sensors-23-05704-t002:** GNSS observation noise.

		Normal		Self-Interference	
**Signal**		**PRNoise** **(mm)**	**CPNoise** **(mm)**	**PRNoise** **(mm)**	**CPNoise** **(mm)**
BDS	B1C	64.73	1.54	65.02	1.49
	B2a	43.48	1.83	46.91	1.92

**Table 3 sensors-23-05704-t003:** Orbit determination scheme.

Scheme	Model
Raw measurement	Dual-frequency combination
Elevation	>10°
Ephemeris	Precise ephemeris (IGS)
Parameter estimation	Extended Kalman filter
LEO dynamics model	Reduced-dynamic approach

**Table 4 sensors-23-05704-t004:** Orbit determination results.

	R (cm)	T (cm)	N (cm)	3D (cm)
BDS	1.05	2.60	1.84	3.35
GNSS	0.89	2.35	1.26	2.82

**Table 5 sensors-23-05704-t005:** Signal quality indexes.

Signal	I/Q Phase Relation	Correlation Loss	S-Curve Bias (1 Chip)
FA/FB	90°	0.3 dB	0.3 ns

**Table 6 sensors-23-05704-t006:** I/Q orthogonality at different elevations for navigation augmentation signal.

Signal	High Elevation	Medium Elevation	Low Elevation
FA	0.38°	0.37°	1.69°
FB	0.32°	0.35°	1.45°

**Table 7 sensors-23-05704-t007:** Correlation loss of navigation augmentation signal.

Signal		High Elevation	Medium Elevation	Low Elevation
FA	In-phase	0.25 dB	0.26 dB	0.34 dB
	Quadrature	0.23 dB	0.29 dB	0.31 dB
FB	In-phase	0.17 dB	0.19 dB	0.29 dB
	Quadrature	0.20 dB	0.22 dB	0.25 dB

## Data Availability

The data that support the findings of this study are available from BeiJing Future Navigation Tech Co., Ltd. and The 29th Research Institute of China Electronic Technology Group Corporation but restrictions apply to the availability of these data, which were used under license for the current study, and so are not publicly available. Data are, however, available from the authors upon reasonable request and with permission of BeiJing Future Navigation Tech Co., Ltd. and The 29th Research Institute of China Electronic Technology Group Corporation.

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
