# Peer review of "Performance Evaluation of CentiSpace Navigation Augmentation Experiment Satellites"

_sensors, 2023, doi:10.3390/s23125704_

Round 1

Reviewer 1 Report

The paper provides timely results on the Centispace LEO-PNT system und should be published soon. The others analyze a few top-level performance indicators which seem to be well according to their data. 

Unfortunately no technical details on the signals or the processing strategy are disclosed. This would be a no-go for a true scientific paper, but since the authors will most likely never get the permission to disclose those details, it should be fine here.

Some minor comments:

* The authors talk about integrity in 2.1.1. and 3.1.1, but they mean "availability", i.e. percentage of data available. Integrity is the probability of gross errors in the signals. This should be changed

* I would be interesting to see C/N0 plots for the case of augmentation signal on and self-interference cancellation technique off to fully appreciate the effect of the self-interference cancellation technique.

Good work!

Only some minor typos are presents. Use of whitepaces after dots should be considered.

Author Response

Thank you for you suggestions.They are very helpful for the paper.Here are my responses:

Response 1:Thanks for your reminding of the definition of integrity.This paper shows the strategy to accept a raw measurement in Figure.1.The raw measurement with gross errors will be deleted for the evaluation of integrity(“Value Check” in the figure).

I have also emphasis this idea in the new paper “the raw measurement should meet the conditions of Figure1”(2.1.1, in red).

Response 2: As described in 2.1.2 “The power of Centispace augmentation signals are more than 100dB higher than that of the normal signals received by the antenna.”.The GNSS receiver can not acquire GNSS signals without self-interference suppression technique.Thus, the C/N0 is zero.

Your suggestion is benefit for readers to appreciate the effect of the self-interference cancellation technique.I have also added the description of C/N0 without self-interference suppression technique “In the event of deactivating the interference suppression algorithm,GNSS receivers will experience signal disengagement.”(3.2,in red)

Reviewer 2 Report

Comments and Suggestions for Authors

1.      The manuscript seems to lack of methodological innovation, which seems more like a technical summary report. It is better to list the specific motivation and contribution of this manuscript to let the readers know more clearly about the specific innovation points of this manuscript in the Introduction.

2.      In Section 2, the authors can provide an overall framework diagram of the performance evaluation methods mentioned in Section 2 to make it easier for readers to understand.

3.      It is better for authors to provide clear and brief explanations and analyses for all existing tables and figures.

4.      In conclusion, it is better to discuss some prospects for future work.

Moderate editing of English language

Author Response

Thank you for you suggestions.They are very helpful for the paper.Here are my responses:

Response 1:This paper presents a comprehensive analysis of LEO practical performance pertaining to both the signal reception and transmission with identical frequency. It is also the first in-orbit verification of the CCST technique in LEO satellites.The performace analyse is helpful to expand the potential for LEO navigation augmentation.

Now,the specific motivation and contribution of this manuscript are listed in the Introduction(in red).

Response 2:Thanks for your good advice.The Framework diagram is now provided in Section 2.

Response 3: Thanks for your reading.I have read the full manuscript carefully,and provided necessary explanations and analyses for tables and figures(in red).

Response 4: Comprehensive investigations and validations pertaining to global integrity enhancement and high-precision positioning will be carried out with the construcion of LEO satellites.Some prospects for future work are now explained in conclusion(in red).

Round 2

Reviewer 2 Report

The paper is much better now.I think this paper meets the acceptance criteria.

English is good.